# Photon deceleration in plasma wakes generates single-cycle relativistic tunable infrared pulses

Zan Nie [1,2], Chih-Hao Pai[1 ✉], Jie Zhang[1], Xiaonan Ning[1], Jianfei Hua [1 ✉], Yunxiao He[1], Yipeng Wu[1], Qianqian Su[1], Shuang Liu[1], Yue Ma[1], Zhi Cheng[1], Wei Lu[1,3 ✉], Hsu-Hsin Chu [4,5], Jyhpyng Wang[4,5,6,7 ✉], Chaojie Zhang[2], Warren B. Mori[2] & Chan Joshi[2]

Availability of relativistically intense, single-cycle, tunable infrared sources will open up new areas of relativistic nonlinear optics of plasmas, impulse IR spectroscopy and pump-probe experiments in the molecular fingerprint region. However, generation of such pulses is still a challenge by current methods. Recently, it has been proposed that time dependent refractive index associated with laser-produced nonlinear wakes in a suitably designed plasma density structure rapidly frequency down-converts photons. The longest wavelength photons slip backwards relative to the evolving laser pulse to form a single-cycle pulse within the nearly evacuated wake cavity. This process is called photon deceleration. Here, we demonstrate this scheme for generating high-power (~100 GW), near single-cycle, wavelength tunable (3–20 µm), infrared pulses using an 810 nm drive laser by tuning the density profile of the plasma. We also demonstrate that these pulses can be used to in-situ probe the transient and nonlinear wakes themselves.

[1] Key Laboratory of Particle and Radiation Imaging of Ministry of Education, Department of Engineering Physics, Tsinghua University, Beijing 100084, China. [2] University of California Los Angeles, Los Angeles, CA 90095, USA. [3] State Key Laboratory of Low Dimensional Quantum Physics, Department of Physics, Tsinghua University, Beijing 100084, China. [4] Department of Physics, National Central University, Jhongli 32001, Taiwan. [5] Center for High Energy and High Field Physics, National Central University, Jhongli 32001, Taiwan. [6] Institute of Atomic and Molecular Sciences, Academia Sinica, Taipei 10617, Taiwan. [7] Department of Physics, National Taiwan University, Taipei 10617, Taiwan. ✉email: chpai@tsinghua.edu.cn; jfhua@tsinghua.edu.cn; weilu@tsinghua.edu.cn; jwang@ltl.iams.sinica.edu.tw

A tunable ultra-short long-wavelength infrared (LWIR, 6–20 μm) laser source is highly desirable in numerous physics, material science, biology, and medicine applications[1–8]. Indeed remarkable progress for generating intense few-cycle mid-infrared (mid-IR) pulses ($\lambda < 4$ μm) has been made in recent years through various optical methods, such as optical parametric amplification[9,10], filamentation[11], and difference-frequency generation[12–14]. However, to extend these methods for producing relativistically intense, single-cycle pulses in the LWIR region is very challenging due to the lack of suitable optical materials with large bandwidth and high damage threshold. Recently, it has been proposed that a wake (density disturbance) generated by an intense pump laser pulse in a properly designed plasma density structure (such as that shown in Fig. 1a) can serve as a new type of nonlinear optical device for generating intense single-cycle broadband LWIR pulses[15]. Physically this happens because of a combination of asymmetric self-phase modulation (ASPM) that mainly produces frequency down-converted photons and group velocity dispersion (GVD) of these photons in the plasma. This combined process is known as photon deceleration in kinetic description of photons in plasma physics[16–18], where strong time-dependent plasma density

(refractive index gradient $\frac{\partial \eta}{\partial \zeta}$ with $\zeta = t - \frac{z}{c}$ being the variable in the speed of light frame) can continuously alter the photon frequency via phase modulation such that the instantaneous frequency is given by $\omega(t) = \omega_0 - \omega_0 \int \frac{\partial \eta}{\partial \zeta} dt$ (or $\frac{1}{\omega} \frac{\partial \omega}{\partial t} = -\frac{\partial \eta}{\partial \zeta}$), and the longer wavelength photons generated by a positive refractive index gradient travel with a smaller group velocity ($v_g(\omega) \simeq c[1 - \frac{\omega_p^2}{2\omega^2}]$) than shorter wavelength photons in the plasma[19–22]. Here $\omega_p = (4\pi n_p e^2/m_e)^{1/2}$ is the plasma frequency, $n_p$ is the plasma density, $m_e$ is the electron mass, and $e$ is the electron charge. When the laser pulse intensity is moderate ($a_0 \sim O(1)$) and the pulse duration is roughly one plasma wavelength, laser photons are frequency downshifted at the front and are upshifted at the back of the pulse as in usual self-phase modulation. Here $a_0 = eE/m\omega c$ is the normalized vector potential of the laser pulse, where $E$ is the electric field of the laser pulse and $c$ is the speed of light. This mechanism has been used for pulse compression[23–25] or as a diagnostic for the generation of wakes in plasmas[26–28].

Recently this concept was extended to explore the possibility of generation of mid-IR pulses using particle-in-cell (PIC) simulations[29,30]. Experimentally, the near-IR pump pulse was

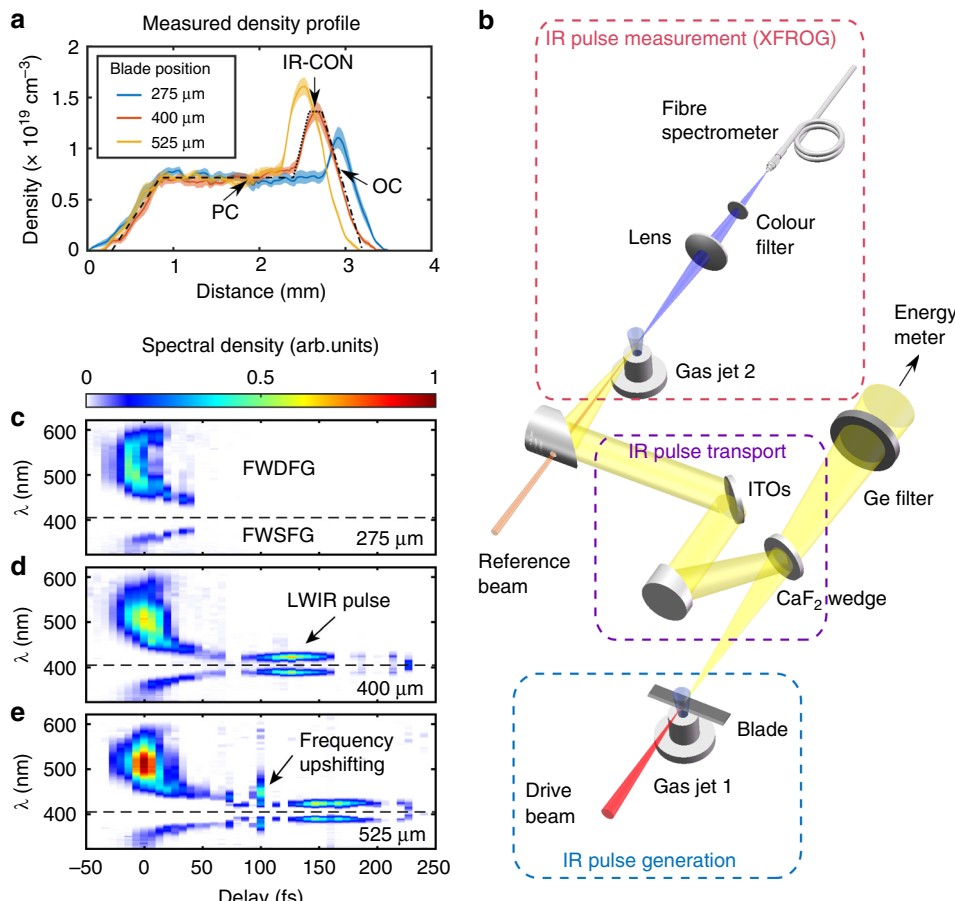

**Fig. 1 Density profiles, experimental setup, and XFROG results. a** The measured on-axis plasma densities (the blurred curves show the RMS spread of the data) at different blade positions (275, 400, and 525 μm from the edge of the opening) into the gas jet. The resulting plasma density profile has three distinct regions—PC: pulse compressor (black dashed line); IR-CON: IR converter (black dotted line); OC: output coupler (black dash-dot line). **b** Schematic of the experimental setup. In the IR pulse generation section, the drive laser (red) is focused onto Gas jet 1 (hydrogen) with a movable blade to interact with the plasma structure (see "Methods"). In the IR pulse transport section, the transmitted light from Gas jet 1 is split by a CaF₂ wedge into two beams: one is sent after being filtered by an uncoated Ge plate to energy meter for IR energy measurement; the other is filtered (ITOs) and coupled to the XFROG device—Gas jet 2 (argon) followed by a spectrometer. **c–e** The measured XFROG traces for the corresponding plasma density profiles shown in (**a**). The black dashed lines mark the position of 405 nm ($\lambda_{ref}/2$). Both four-wave difference-frequency and sum-frequency generated spectra (FWDFG and FWSFG, respectively) can be seen.

broadened to mid-IR wavelengths using uniform density plasmas[31–33]. However, none of the previous works to date have succeeded in extending the concept of photon deceleration in plasma wakes to generate near single-cycle, tunable, and relativistic pulses in the LWIR region. Here, we experimentally demonstrate the generation of such pulses containing millijoules of energy in the 3–20 μm wavelength range by utilizing the interaction between an intense laser pulse and the nonlinear wake it generates in a specially prepared plasma density structure.

## Results

**Concept and experimental realization.** The plasma density structure used in the experiments is a low-density platform with a sharp density upramp (Fig. 1a). We use a supersonic hydrogen gas jet target with an insertable blade to shock-induce a density spike[34–36] in the gas flow to produce the necessary density profile of the gas (Fig. 1b). When an ultra-short ($36 \pm 2$ fs, FWHM), energetic ($580 \pm 9$ mJ), 810 nm wavelength drive laser pulse[37] passes through, it produces both a fully ionized plasma via field-induced ionization[38] and a highly nonlinear wake[39] throughout the structure. Briefly, the ponderomotive force of the focused laser pulse is large enough to expel nearly all the plasma electrons leaving behind the more massive ions. These electrons are however attracted back by the Coulomb force of the ions forming a bubble-like wake cavity where the ions are encapsulated by a thin sheath of plasma electrons. The strong time-dependent plasma density gradients formed during the expulsion of the plasma electrons phase modulate the laser pulse downshifting the instantaneous frequency of the photons as explained earlier while the wavelength-dependent group velocity disperses these photons.

The measured density profiles of the plasma structure for three different settings of the insertion position of the blade in the gas flow emanating from the supersonic gas nozzle are shown in Fig. 1a. The first section of the plasma structure is called the pulse compressor (PC). The PC is a few-millimeter-long relatively low-density section (where the wake dimension is on the order of laser pulse length) that first chirps and then self-compresses much of the drive laser pulse from typically 40 to 10 fs. This is caused by the combined effect of gentle photon deceleration at the front where the plasma electrons are being blown out by the laser pulse followed by photon acceleration at the back of the pulse and negative GVD within the wake analogous to chirping and compression in a fiber with nonlinear refractive index $n_2$ and GVD having opposite signs. The second section is called the IR converter (IR-CON). The IR-CON is a shorter, higher-density section where large refractive index gradients formed by the compressed ultra-short pulse (far shorter than the wake cavity dimension) lead to rapid and efficient photon deceleration (by ASPM). The longer wavelength components recede rapidly, phase lock, and eventually reside in the main body of the wake to generate a near single-cycle LWIR pulse. Finally, this LWIR pulse is coupled from plasma to vacuum by the third section called an output coupler (OC). The OC is a plasma downramp used to further tune the carrier wavelength of the LWIR pulse and transport it out of the plasma structure with little attenuation by gradually elongating the wake[15]. By using plasma as a nonlinear medium we circumvent the damage considerations that limit the power densities in other optical methods[9,10,12–14] to generate relativistically intense, tunable, near single-cycle LWIR pulses.

By controlling the peak density of the IR-CON section of the plasma structure, we have generated tunable IR pulses in the wavelength range of 3–20 μm. The IR pulse is characterized by cross-correlating it with a second synchronized 810 nm reference pulse, with a known intensity and phase (characterized using a separate Wizzler device, see Supplementary Fig. 5), through cross-correlation frequency-resolved optical gating (XFROG)[40] based on four-wave mixing (FWM) as shown in Fig. 1b. In the XFROG measurement, the residual drive pulse is filtered out so that only photons in the wavelength range > 1.5 μm are transmitted and detected.

**LWIR pulse optimization.** The plasma density profile is crucial for LWIR pulse generation. We manipulate the peak neutral density of hydrogen gas in the IR-CON region without significantly affecting either the density of the PC or the scale-length of the OC by varying the insertion position of the blade relative to the edge of the gas jet from 275 to 525 μm (Fig. 1a). The ultra-short laser pulse then produces a plasma with a density profile proportional to the neutral density profile through field-induced ionization[38]. Three examples of the measured plasma density profiles (Fig. 1a) and the measured XFROG traces for these profiles are shown in Fig. 1c–e. The XFROG traces show the spectra of four-wave sum-frequency generation (FWSFG) and four-wave difference-frequency generation (FWDFG) as functions of time. The entire cross-correlation plot is generated by varying the delay between the IR pulse and the reference 810 nm pulse using approximately 240 shots, 5 per particular delay between the IR and the reference pulse. Both FWDFG (where $\omega_{ref} + \omega_{ref} - \omega_{IR} = \omega_{FWDFG}$) and FWSFG (where $\omega_{ref} + \omega_{ref} + \omega_{IR} = \omega_{FWSFG}$) signals are observed as expected. Here $\omega_{ref}$, $\omega_{IR}$, $\omega_{FWDFG}$, and $\omega_{FWSFG}$ are the angular frequencies of the reference, the wake-generated IR, the difference and sum frequency radiation generated by XFROG, respectively. Since both FWDFG and FWSFG processes involve the same input optical fields, these traces are symmetric with respect to the $\lambda_{ref}/2 = 405$ nm axis (black dashed line). Subtracting the expression for FWDFG from the FWSFG expression and substituting the measured wavelengths observed for $\lambda_{FWSFG}(t)$ and $\lambda_{FWDFG}(t)$ (see Fig. 1c–e) one directly obtains $\lambda_{IR}(t)$. However, we use the FWDFG signal for XFROG retrieval[41] to get both the intensity and phase information of the IR pulse throughout this paper.

When the insertion position of the blade is at 275 μm (Fig. 1c, a —blue curve), the IR-CON section of the plasma does not have high enough density to generate the LWIR pulse—the FWDFG spectrum spans roughly from 600 to 435 nm corresponding to an IR spectrum that spans from 1.5 to 6 μm (MIR). This is the usual MIR spectrum reported in experiments on wake generation in uniform plasmas[31]. When the insertion position of the blade is moved to 400 μm, the peak density of the IR-CON region is now sufficient (Fig. 1a—red curve) to give rise to a very rapid downshifting of photons and thereby generate the LWIR component (the 423 nm peak in FWDFG XFROG trace corresponding to a peak IR wavelength of 9.4 μm) as seen in Fig. 1d (here the LWIR signal is seen to follow the MIR pulse due to its slower group velocity in the plasma). With a further increase of the insertion position of the blade to 525 μm (Fig. 1a— yellow curve), the LWIR pulse is seen to be stretched longer in time (Fig. 1e) and in addition a frequency upshifted signal now appears at a delay of 100 fs. By retrieving many such XFROG traces, we find that the shortest LWIR pulses are produced when the blade is positioned at 400 μm. In this case, the RMS difference between the measured (Fig. 2a) and retrieved (Fig. 2b) XFROG traces is 0.9%. The temporal (spectral) intensity and phase of the IR pulse are shown in Fig. 2c, d. We focus on the LWIR component (the right dashed box in Fig. 2b) of the measured IR pulse and retrieve it alone (Fig. 2e, f). This gives a pulse duration of 32.0 fs (FWHM) and a central wavelength of 9.4 μm. This pulse duration is close to the period of a 9.4 μm wavelength (31.3 fs), indicating the generation of a near single-cycle pulse produced by phase locking of the higher and lower wavelength photons

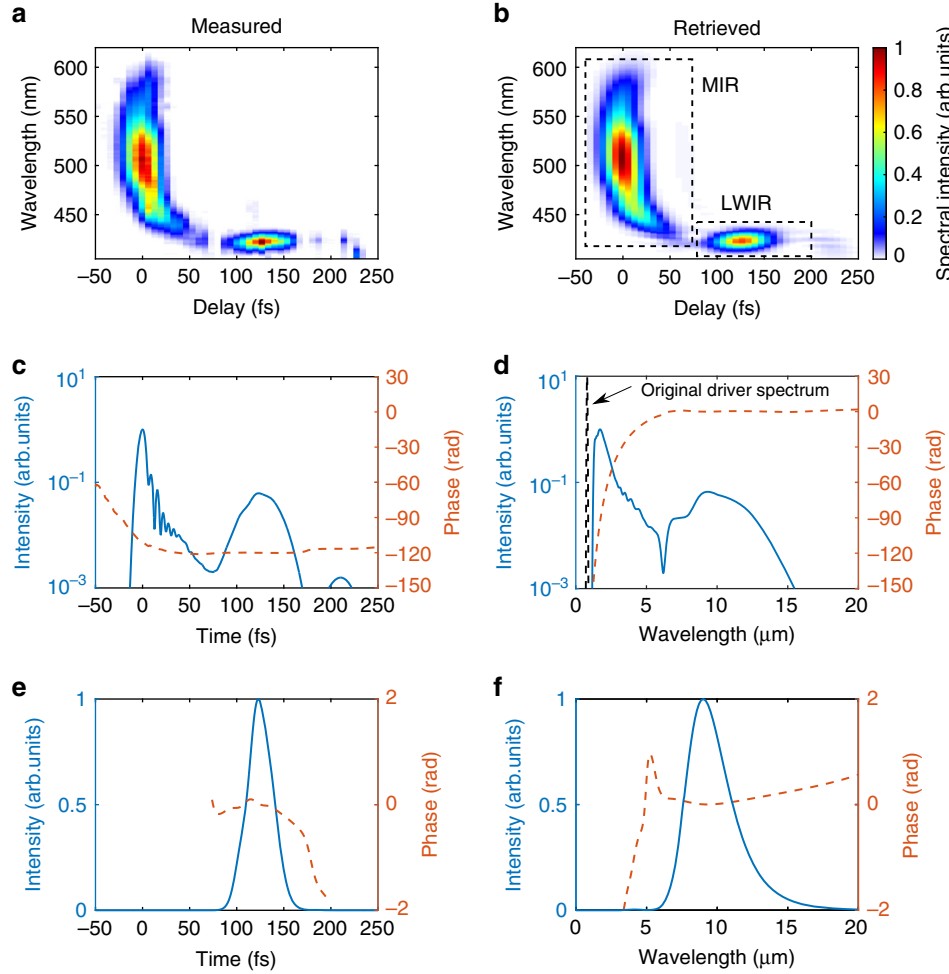

**Fig. 2 Retrieval of the XFROG trace. a**, **b** Measured and retrieved XFROG traces. The two dashed boxes in (**b**) are the FWDFG signals generated by MIR (1.5–6 μm) and LWIR (6–20 μm) components, respectively. **c**, **d** Retrieved IR temporal and spectral intensity and phase. The intensities are shown on a logarithmic scale. The dashed black line in (**d**) shows the original spectrum of the 810 nm drive laser. **e**, **f** Retrieved LWIR temporal and spectral intensity and phase. The gentle quadratic phase variation shown in (**e**) and (**f**) is consistent with a small linear chirp seen in (**a**).

around this carrier frequency. This is confirmed by the nearly flat phase of the retrieved LWIR pulse that suggests a near-transform-limited ($\Delta\nu\Delta t = 0.446$) pulse with a small linear chirp generated in this process.

**LWIR energy**. The IR energy and XFROG signal (at a particular time) are measured simultaneously on every shot during the experiment. The measured mean IR energy for the case shown in Fig. 2 is 133 ± 42 μJ. By correcting for the transport efficiency and FWM efficiency (see Supplementary Note 2), the estimated mean LWIR energy (in the range 6–20 μm) generated at Gas jet 1 is no lower than 3.4 ± 1.1 mJ, corresponding to a peak power of no lower than 106 GW, and a conversion efficiency of no lower than 0.6%. This efficiency can be further improved by optimizing the driving laser's beam quality and the plasma density structure (see Supplementary Note 3). Due to the conservation of wave action[22,42,43] (or photon number) in this photon deceleration process, the ideal conversion efficiency is limited by the LWIR to pump frequency ratio (the quantum efficiency, ~8% in this case). Accordingly, the normalized vector potential $a_0$ of the LWIR pulse at the exit of Gas jet 1 is about 1.53 ± 0.25. To estimate $a_0$, the measured pulse energy and pulse duration are used with the assumption that the spot sizes of the IR pulses are close to the transverse wake size (verified by 3D PIC simulations shown below). This confirms that the intensity of the LWIR pulse

reaches a relativistic level (when $a_0 \sim O(1)$ electrons oscillate in the laser field near the speed of light).

**Wavelength tunability**. The central wavelengths of the IR pulses are tuned from 3.2 to 20.0 μm by varying several experimental parameters (the gas pressure, the blade position relative to the gas jet, and the driving laser energy). In Fig. 3, the measured IR pulse energy for different wavelength cases is plotted together with the estimated normalized vector potential $a_0$ of the IR pulses at the exit of the plasma structure. The detailed XFROG data for each case is shown in Supplementary Figs. 11–14. In all cases, the IR energy is in several millijoule level, and the $a_0$ is estimated to be at relativistic level ($a_0 \gtrsim 1$) at the exit of the plasma structure. The variations in energy or pulse duration between different cases are not due to any plasma instability, but only due to the variation of multiple parameters as mentioned above. In order to system-atically tune the LWIR wavelength and reduce the fluctuations in energy or pulse duration, precise and reproducible control of the plasma density profile is required[15].

**Discussion**
The time evolution of the IR pulses inferred from the XFROG traces has already given insight into the physics of highly tran-sient (few picoseconds), microscopic (tens of micron size) and highly-nonlinear, speed of light wakes. Now we show the

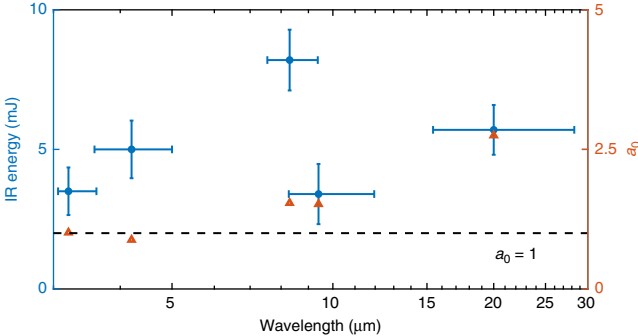

**Fig. 3 Wavelength tunability.** The data point with central wavelength of 9.4 μm corresponds to that shown in Fig. 2 and the detailed XFROG data for other four cases are shown in Supplementary Figs. 11–14. The data points for IR energy are averaged over 240 shots. The wavelengths are shown on a logarithmic scale. The horizontal bars represent the wavelength ranges (FWHM) of the accumulated IR energy. The vertical bars represent the standard deviation of IR energy. The solid orange triangles correspond to the estimated normalized vector potential $a_0$. The black dashed line marks the value of $a_0 = 1$.

application of the generated IR pulses as an in-situ probe of the nonlinear wakes. To illustrate this, the Wigner spectrograms (frequency vs time) of the IR pulses retrieved from XFROG traces in Fig. 1d, e are presented in Fig. 4a, b, respectively. As we increased the peak density by moving the blade position from 400 to 525 μm (Fig. 4a, b) the latter case first shows strong photon deceleration ($0 < t(\text{fs}) < 100$) followed by photon acceleration at a delay of 100 fs. This is then followed by a second even longer wavelength IR component (center at 160 fs), indicating of this second IR component has leaked through the first wake bucket into the second and is encased there. Simulations using the 3D PIC code OSIRIS[44,45] (with the same density profile as in the experiment) qualitatively verify the above scenario. The snapshots of the simulated Wigner spectrograms (frequency vs time) of the on-axis laser electric fields corresponding to the experimental cases in Fig. 4a, b are displayed in Fig. 4c, d, respectively. The corresponding snapshots of the plasma density (wake) and laser electric field are shown in Fig. 4e, f, presenting two very different wakefield structures. In both cases, continuous self-injection of electrons happens. In Fig. 4e, the self-injection of electrons leads to gradual elongation of the first bucket[46,47] but the resultant beam loading and the ponderomotive force of the LWIR pulse are not strong enough to force the back of the bucket to stay open. Therefore, the LWIR pulse remains in the first bucket and there is no frequency upshifting component observed towards the end of the wake. However, as seen in Fig. 4f, when the density of the IR-CON region is increased further, even stronger self-injection of electrons and the ponderomotive force of the LWIR pulse trapped inside the first cavity pushes the returning plasma electrons in the sheath at the very back and forces the first bucket to open wide. At this moment, the majority of the LWIR radiation leaks through this opening into the second bucket. Once the majority of the LWIR radiation has passed, the opening tends to shrink again due to the decrease of the ponderomotive force. Those LWIR radiation right at the location of the shrinking opening will be frequency upshifted. In this way, a double peanut-shaped wake is formed and this wake structure feature is marked by the frequency-upshifted component. In the simulations, one can see this frequency-upshifted component of the LWIR pulse at the end of the partially closed first bubble where the negative refractive index gradient $-\text{d}\eta/\text{d}\zeta$ is positive (orange curve in Fig. 4d), which is consistent with the frequency-upshifting component in the experimental result (in Fig. 4b at $t = 100$ fs). In other words, the

highly nonlinear, transient wakefield structures are mapped into the transient spectra of the IR pulses. This is but one example of applications of such LWIR pulses for unraveling the nonlinear optics of relativistic plasmas.

There is a good qualitative agreement between the experimental results (Fig. 4a, b) and the simulation results (Fig. 4c–f) regarding the spectrum, pulse duration, and overall energy of the LWIR pulse (more details are shown in Supplementary Note 3). The one obvious discrepancy is the delay between the MIR pulse and LWIR pulse. For 400 μm case, the experimental delay is about 120 fs, but the simulated delay is about 50 fs. For 525 μm case, the experimental delay is about 160 fs, but the simulated delay is about 110 fs. One possible reason is that the real plasma density profiles have longer low-density tails ($<7 \times 10^{17}$ cm$^{-3}$) that are below the noise level of our density diagnostics. Low but long density tails may lead to a larger delay between LWIR pulse and MIR pulse due to significantly large group velocity difference between two wavelengths.

In conclusion, the generation of tunable, relativistic, near single-cycle IR pulses, and their applications to self-probing of the highly nonlinear plasma wake has been demonstrated using a tailored plasma structure. Given that a few TW class, femtoseconds drive lasers in the near-IR are now commonplace, one expects this technique to be adopted in many laboratories to give intense, tunable LWIR pulses. When scaling this scheme to higher energies using >100 TW lasers, spatio-temporal effects[48,49] may affect the laser propagation and wake generation in plasmas and hence the generation of the LWIR pulse. At relativistic intensities afforded by such sources it will now be possible to study electron injection[50–55], beam loading[56–58], spin polarization[59], and emittance[60] preservation in large diameter wakes generated in plasmas on one hand and at lower intensities study high harmonic generation[3–5] and resonant or non-resonant nonlinear interactions in gases, solids, or biological systems[1,2,6–8] on the other.

## Methods

**Experimental setup.** A schematic of the experimental setup is shown in Fig. 1b. The 810 nm, 36 ± 2 fs (FWHM) drive-laser pulse[37] (red) with a contrast of $10^8$ containing 580 ± 9 mJ energy is focused onto the low-density side of Gas jet 1 (hydrogen) to a spot size $w_0 = 13.5 \pm 1.0$ μm. Approximately 60% of the laser energy is within the beam waist ($1/e^2$ of intensity), corresponding to an estimated Strehl ratio of 0.7. The necessary density structure is produced by inserting a movable blade placed just above the opening of a 3 mm diameter supersonic gas nozzle. The blade partially interrupts the gas flow and induces a shock in the gas density profile above it. This results in a fluid density profile that has a few-millimeter-long low-density region at the beginning, that transits into a shock-induced higher-density but shorter (sub-millimeter) region (Fig. 1a). An uncoated calcium fluoride (CaF$_2$) wedge is used to split the laser pulse exiting Gas jet 1 into two beams. The first passes through the wedge and is filtered using an uncoated germanium (Ge) plate to cut any residual 810 nm light. The transmitted light in the wavelength range between 2 and 20 μm is used for total IR energy measurement.

The reflected portion of the pulse is used for the characterization of the IR pulse as follows: First, the residual drive pulse ($\lambda < 1.5$ μm) in the reflected beam are filtered out by six IR beam splitters coated with indium tin oxide (ITO) so that only the IR photons with $\lambda > 1.5$ μm can be transported for characterization. Second, a synchronized second 810 nm reference pulse (brown) with known temporal amplitude and phase is used for performing a XFROG measurement[40]. The IR pulse and reference pulse are focused separately using off-axis parabolas and overlap with each other at Gas jet 2 (argon). These two pulses interact with each other through FWM in argon and generate the XFROG signals (blue) that are measured by a fiber spectrometer after a bandpass color filter. The FWSFG signal is weaker than the FWDFG signal because the former process is less efficient than the latter in most of the spectral range due to phase mismatch and beam mode mismatch[61] (see Supplementary Note 2). Besides, the transmission of the bandpass filter below 360 nm decreases significantly, which results in a weak FWSFG signal below 360 nm.

**Characterization of plasma density structures.** The plasma density structure is produced by a supersonic round nozzle with a blade covering a portion of the gas jet. We combine online and offline measurements to obtain the plasma density profile. In the online measurement, we use interferometry to acquire the plateau plasma density ($7.2 \times 10^{18}$ cm$^{-3}$) in the PC section, since the plasma densities in

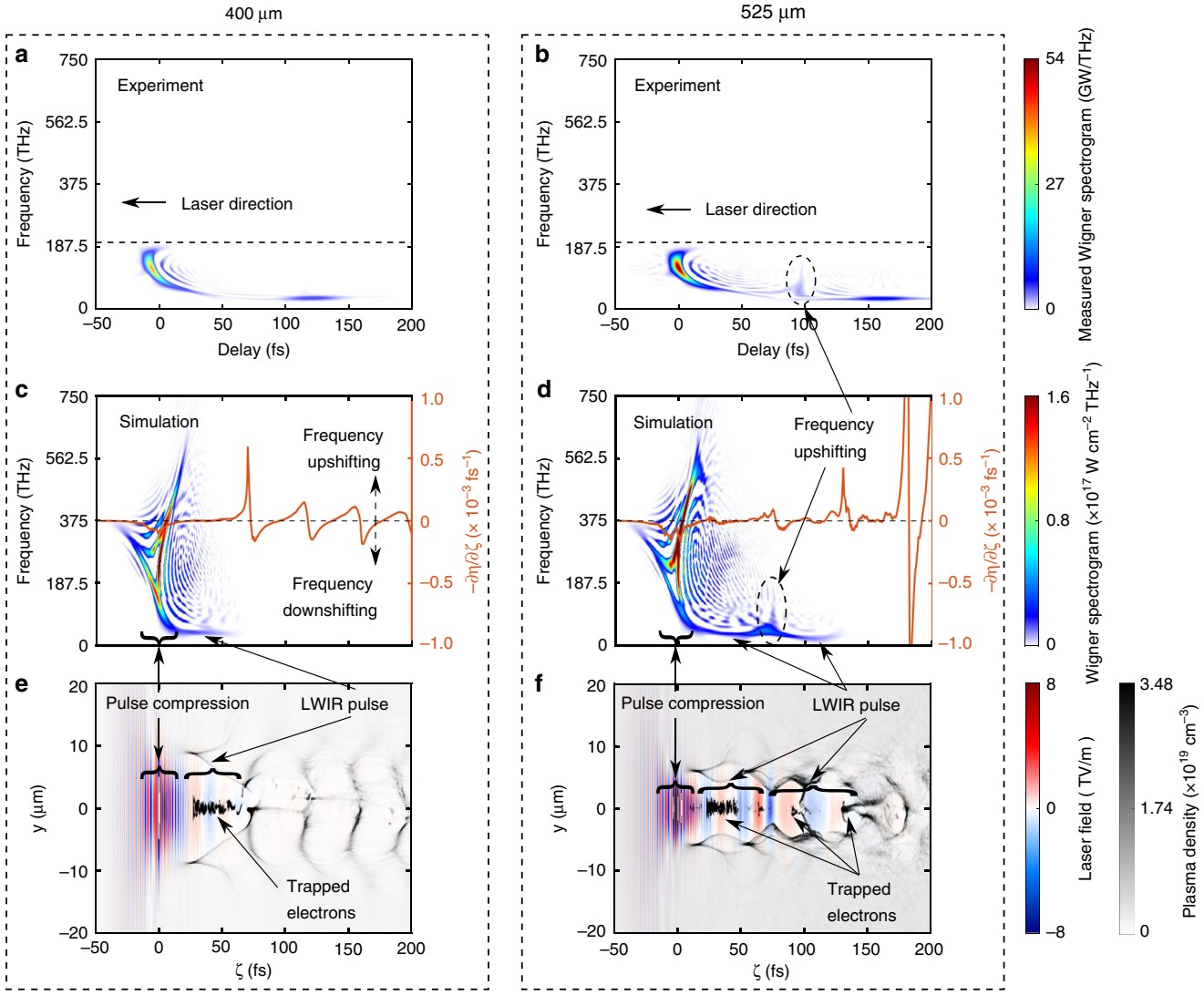

**Fig. 4 Comparison of measured and simulated Wigner spectrograms of generated IR pulses. a**, **b** Wigner spectrograms of measured IR pulses when the blade is at 400 and 525 μm, respectively. The dashed lines represent the upper limit of the measurable IR frequency range due to the optics used to transport the LWIR radiation to the XFROG device. The IR components above 200 THz ($\lambda < 1.5\,\mu m$) are filtered out to remove strong residual drive pulse. **c**, **d** Simulated Wigner spectrograms of one snapshot of the evolved on-axis laser field and the negative refractive index gradient $-\partial\eta/\partial\zeta$ (orange line) in the plasma downramp (the OC section) for the cases in (**a**) and (**b**), respectively. The local frequency of the photons is downshifted (upshifted) when this negative gradient is negative (positive). **e**, **f** The corresponding snapshot of plasma density (wake) and the laser electric field for the cases in (**c**) and (**d**), respectively. Strong photon frequency down-shifting occurs at the front of the self-compressed pulse generating a LWIR pulse that recedes by GVD into the first wake cavity (**c**) and partially leaks into the second wake cavity in the case of (**d**).

the IR-CON and OC sections are non-axisymmetric and thus they cannot be retrieved by the Abel inversion method. In the offline measurement, hydrogen is replaced with argon to get a larger refractive index change. A wavefront sensor (SID-4, PHASICS) camera is used to measure the phase difference of the neutral argon gas at a particular backing pressure at ten different angles. Then two-dimensional density profiles can be reconstructed by using a tomographic reconstruction algorithm. It has been verified by fluid simulations that the density profiles of different gases are similar for the same gas pressure. Therefore, the complete plasma density profile is obtained by multiplying the measured offline density profile (normalized to the plateau density in the PC section) with the measured online plateau density in the PC section ($7.2 \times 10^{18}\,cm^{-3}$). (see Supplementary Note 1).

**XFROG measurement**. In contrast to standard XFROG measurement usually implemented in a nonlinear crystal[40], XFROG measurement here is based on FWM in argon[11]. Gaseous media are intrinsically broadband due to low dispersion over a large spectral range, which helps to avoid the stretching of pulses during XFROG measurement and to achieve broadband FWM phase matching. Therefore, XFROG in gases enables the characterization of ultra-short pulses with ultra-broadband spectra. In our experiment, XFROG traces are obtained by scanning the delay

between the IR pulse and the reference pulse. At each delay, five shots of data are averaged to improve signal-to-noise ratio. The step of the scanning stage is 1 μm (optical path of 2 μm), corresponding to 6.6 fs in delay. For XFROG retrieval, the intensity and phase information of the reference pulse is obtained (60 shots average, see Supplementary Fig. 5) by self-referenced spectral interferometry[62] with a commercial product Wizzler (Fastlite). The retrieval of temporal/spectral intensity and phase follows standard XFROG retrieval algorithm[41].

**PIC simulations**. The 3D PIC simulations were carried out using the code OSIRIS[44,45] in Cartesian coordinates with a window moving at the speed of light. The z-axis was defined to be the drive laser propagating direction. The simulation window had a dimension of $76 \times 76 \times 92\,\mu m^3$ with $400 \times 400 \times 3600$ cells in the x, y, and z directions, respectively. This corresponded to cell sizes of $\Delta x = \Delta y = 1.33 k_0^{-1}$ and $\Delta z = 0.20 k_0^{-1}$ (where $k_0 = 2\pi\lambda_0^{-1}$ is the laser wavevector and $\lambda_0 = 800\,nm$). The number of macro-particles per cell was 8.

## Data availability
The data that support the findings of this study are available from the corresponding author on reasonable request.

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

## Acknowledgements
The authors thank Yau-Hsin Hsieh, Shih-Chi Kao, and Yao-Li Liu for helping with experiments. The authors also thank Professor Takao Fuji for a fruitful discussion of the XFROG measurement in gases. This work was supported by the National Natural Science Foundation of China (NSFC) grants (Nos. 11535006, 11991071, 11875175, 11775125, and 11425521); the Air Force Office of Scientific Research (AFOSR) under Award Number FA9550-16-1-0139 DEF, the Office of Naval Research (ONR) Multidisciplinary University Research Initiative (MURI) (4-442521-JC-22891), the U.S. Department of Energy grant DE-SC001006; and the Ministry of Science and Technology of Taiwan under Grant No. MOST-105-2112-M-001-005-M3. The simulations were performed on Sunway Taihu-Light and the resources of the National Energy Research Scientific Computing Center.

## Author contributions
W.L., J.W., and C.J. conceived and supervised the project. C.-H.P., Z.N., Q.S., S.L., Y.M., and Z.C. developed the XFROG method for IR pulse measurement. J.H. led the development of plasma source. Z.N. and Y.H. performed the plasma density measurement. Z.N., J.Z., X.N., C.-H.P., H.-H.C., and J.H. carried out the LWIR pulse generation and measurement experiments. Z.N. performed the simulations. C.J., W.L., and Z.N. wrote the paper. C.J., W.L., J.W., Z.N., C.-H.P., J.H., Y.W., C.Z., and W.B.M. discussed the results and commented on the paper.

## Competing interests
The authors declare no competing interests.
