## [Peer Review File · Nature Communications]

Reviewers' comments:

Reviewer #1 (Remarks to the Author):

The manuscript "photon deceleration in plasma wakes generates single cycle relativistic, tunable infrared pulses" describes experiments, supported by simulations and theory that show the generation of relatively long wavelength, intense, single cycle infrared radiation pulses. The pulses are generated by the down-conversion of mid infrared (initially narrow 800 nm pulses) light in a dynamic medium – the electron density wake in a plasma that follows an intense laser pulse. The work is carefully carried out, well documented, and novel. I believe it is worthy of publication.

My only concern is that I worry how accessible it will be to a broad scientific community. The basic process of photon deceleration in a nonlinear plasma wake is well known in the small laser-plasma community. However, I suspect that in the general optics community it is novel, and would require more initial explanation and illustration than offered here. That is not to say that the present work does not push the field forward. The authors' implementation of a tailored density profile and enhancement of the efficiency is an important contribution. (They should mention Manley-Rowe constraints.) I am not sure the cleverness of their scheme will come across to the non-plasma-specialist readers.

Perhaps it is not my place to offer advice. But if I were to write this paper, I would start with a simulation showing step by step the evolution of the Wigner function as the pulse and wake traveled through the density profile. Then all readers would understand the processes involved. Subsequently, I would describe the experimental realization and illustrate the points of agreement between the measurements and simulations. This evolution of the pulse and wake as a function of distance cannot be easily illustrated based on the experimental results because only the pulse exiting the interaction region is available for diagnosis.

In conclusion this is excellent work. I feel a better presentation would make it more accessible to a broad readership.

Reviewer #2 (Remarks to the Author):

The manuscript "Photon deceleration in plasma wakes generates single-cycle, relativistic, tunable infrared pulses" reports on experimental results on intense few cycle long-wavelength infrared (LWIR) pulses produced by photon "deceleration" of a near-IR, ultrahigh intensity pulse in a plasma wake. The concept was proposed and illustrated via simulations in a previous Nature Photon. paper by the same group of authors. Since the experimental results basically demonstrate the LWIR pulse generation mechanism, their publication in Nature Comm. as a follow-up of the Nature Photon. paper appears to be natural. However, before publication I suggest some minor clarifications and amendments to the manuscript.

1) Fig.3 shows the LWIR pulse energy and corresponding dimensionless "relativistic" parameter a_0 as a function of the central wavelength, in order to demonstrate the wavelength tunability. If I am not misunderstanding the plot, it is surprising that a slight change in the wavelength around 9 microns (actually within the spectral range represented by horizontal errorbars) produces a near 3-times jump in the pulse energy, and even more surprising that this does not produce a change in a_0 ; does this mean that the energy change is due to a proportional increase in the LWIR pulse duration? If so, what are the possible reasons and the implications on the actual tunability and stability of the LWIR source?

2) In order to explain the quantitative difference between experimental and simulation results, the authors notice that "the real plasma density profiles likely have longer low-density tails than the

measured and simulated plasma density profiles." Does this mean that both the "online" and "offline" method are unable to measure the plasma density in such low-density tails, and why?

3) Nearly a quarter of a century after the first publication of 3D particle-in-cell simulations of laser-plasma interactions, I am still horrified by the use of only 2 particles per cell. What is the impact of such low resolution of the phase space? How can the convergence of the results be ensured? As a minor suggestion, the same temporal window should be used in Fig.4a-b and Fig.4c.f.

Reviewer #3 (Remarks to the Author):

Generating high energy mid-infrared (MIR) pulses is a topic of interest from multidisciplinary point of view. That is why most of the upcoming laser facilities are incorporating MIR lasers in their user package (See J. Phys. B: At. Mol. Opt. Phys. 50 132002 (2017) for example). In the relativistic domain accessibility to a few cycle intense pulses at several microns central wavelength would be helpful for addressing many fundamental questions. For example at present, field driven phenomena at high intensity relativistic surface physics (Phys. Rev. Lett. 110, 175001 (2013)) is fundamentally limited by the control-ability of plasma surface density gradient (L) with respect to the interaction wavelength (λ). This can be overcome when strong few cycle MIR/IR pulses would be available such that $\lambda \gg L$. In the manuscript "Photon deceleration in plasma wakes generates single-cycle, relativistic, tunable infrared pulses" the authors make a step towards generating such pulses. Thus the results are very interesting, significant and relevant. The authors use cross disciplinary techniques and simple ideas to demonstrate Strong MIR/IR pulses. The results are sound and attractive, however I have several questions that the authors need to address to improve the manuscript. Below I summarize my comments/recommendations:

1. In the abstract (last sentence) they claim demonstration of application of 'these pulses' as an in situ probe of the nonlinear wakes themselves. Elaboration on this part latter in the manuscript missed to capture my attention. They either need to remove these claim or clearly bring out where it is demonstrated in the article.

2. Analysis of the beam profile of the generated MIR/IR beam is missing. This kind of limits their claim of millijoules of energy. This part needs to be strengthened or clarified. Otherwise figure 3 gives the impression of extrapolation.

3. In the title the authors claim and emphasize the single cycle nature of the pulse and in lines 188-191 they discuss observation of almost transform limited pulses. Few points are not brought out with clarity. Physics-wise what is the correlation between the phase properties of the driving pulse and the generated MIR/IR pulse? Why is it transform limited? What is the effect of chirp on the generated long wavelength pulses? How do the phase properties of the generated secondary long wavelength pulse depend up on the plasma stages? These are essential questions that would shed light on the control and utility of the generated pulses. I would recommend that the authors to address these points in more detail in the manuscript.

4. In the time domain plots (figure 2 for example), it would be better to give an impression of the field and include a discussion of the CEP of the long wavelength part, and its correlation with plasma conditions and drive laser conditions. If the pulse is single cycle, such effects would be of paramount importance, hence mandatory in these types of work.

5. Spatiotemporal effects, which are usually part of big lasers (Appl. Phys. Lett. 104, 054103 (2014); Optics Letters 39, 4687(2014)), might play a role in the generation and observation, specially considering the propagation in an asymmetric shock density profile. A clear comment/discussion would be beneficial for the readership.

In addition, in order for the pulses to be used for subsequent experiments it would be important to know the effects of beam pointing fluctuations, and how stable the pulse energy, pulse CEP for the long wavelength pulses. I would suggest that the authors include such plots giving indication how such a source compare with existing sources.

With these points I believe that the quality of the work would improve substantially and be suitable for publication for publication in Nature Communications.

Best wishes,
Subhendu KAHALY
Division Head
Secondary Sources Division

ELI-HU Non-Profit Ltd.
HQ: Dugonics tér 13. Szeged H-6720
Office: 3 Wolfgang Sandner utca, Szeged 6728

Response to Referees

Reviewer #1 (Remarks to the Author):

The manuscript “photon deceleration in plasma wakes generates single cycle relativistic, tunable infrared pulses” describes experiments, supported by simulations and theory that show the generation of relatively long wavelength, intense, single cycle infrared radiation pulses. The pulses are generated by the down-conversion of mid infrared (initially narrow 800 nm pulses) light in a dynamic medium – the electron density wake in a plasma that follows an intense laser pulse. The work is carefully carried out, well documented, and novel. I believe it is worthy of publication.

My only concern is that I worry how accessible it will be to a broad scientific community. The basic process of photon deceleration in a nonlinear plasma wake is well known in the small laser-plasma community. However, I suspect that in the general optics community it is novel, and would require more initial explanation and illustration than offered here. That is not to say that the present work does not push the field forward. The authors’ implementation of a tailored density profile and enhancement of the efficiency is an important contribution. (They should mention Manley-Rowe constraints.) I am not sure the cleverness of their scheme will come across to the non-plasma-specialist readers.

We thank the referee for the supportive comments. Manley-Rowe relations are wave-action conservation laws relating to nonlinear processes of multi-wave interactions. In plasmas, Manley-Rowe relations are applicable to parametric instabilities such as stimulated Raman and Brillouin scattering just as in nonlinear optics. In the case of continuous asymmetric self-phase modulation such as we have here, similar wave-action conservation law still exists (Ref. 36 in the manuscript). In other words, the photon number is conserved even though the photon frequency is downshifted. That’s one of the reasons why this process is called “photon deceleration”. Therefore, the conversion efficiency is limited by the LWIR to pump frequency ratio (the quantum efficiency, ~8% in our case) assuming that all pump photons are frequency downshifted to a specific LWIR frequency. In reality, the typical conversion efficiency is usually a few times lower than the quantum efficiency. For instance, in well-optimized simulation cases (Ref. 10 in the manuscript), about 25% of pump photons can be frequency downshifted to the LWIR range we care about (~10 μm), leading to an energy conversion efficiency of ~2%. We have now added this description in the **LWIR energy** section of the text: “Due to the conservation of wave action³⁶ (or photon number) in this photon deceleration process, the ideal conversion efficiency is limited by the LWIR to pump frequency ratio (the quantum efficiency, ~8% in this case).”

Perhaps it is not my place to offer advice. But if I were to write this paper, I would start with a simulation showing step by step the evolution of the Wigner function as the pulse and wake traveled through the density profile. Then all readers would understand the processes involved. Subsequently, I would describe the experimental realization and illustrate the points of agreement between the measurements and simulations. This evolution of the pulse and wake as a function of distance cannot be easily illustrated based on the experimental results because only the pulse exiting the interaction region is available for diagnosis.

We agree with the referee's suggestion on how to explain a new concept aiming at a broad scientific community by showing simulation results step-by-step. However, since the involved physics of this method has been discussed in our previous theoretical paper (Ref. 10 in the manuscript), we mainly focus on the experimental demonstration after a brief description of photon deceleration at the beginning in this paper.

In conclusion this is excellent work. I feel a better presentation would make it more accessible to a broad readership.

We thank the referee for the supportive comments. We have improved the manuscript according to all the referees' comments.

Reviewer #2 (Remarks to the Author):

The manuscript "Photon deceleration in plasma wakes generates single-cycle, relativistic, tunable infrared pulses" reports on experimental results on intense few cycle long-wavelength infrared (LWIR) pulses produced by photon "deceleration" of a near-IR, ultrahigh intensity pulse in a plasma wake. The concept was proposed and illustrated via simulations in a previous Nature Photon. paper by the same group of authors. Since the experimental results basically demonstrate the LWIR pulse generation mechanism, **their publication in Nature Comm. as a follow-up of the Nature Photon. paper appears to be natural.** However, before publication I suggest some minor clarifications and amendments to the manuscript.

1) Fig.3 shows the LWIR pulse energy and corresponding dimensionless "relativistic" parameter a_0 as a function of the central wavelength, in order to demonstrate the wavelength tunability. If I am not misunderstanding the plot, it is surprising that a slight change in the wavelength around 9 microns (actually within the spectral range represented by horizontal errorbars) produces a near 3-times jump in the pulse energy, and even more surprising that this does not produce a change in a_0 ; does this mean that the energy change is due to a proportional increase in the LWIR pulse duration? If so,

what are the possible reasons and the implications on the actual tunability and stability of the LWIR source?

The purpose of Fig.3 is to show the wavelength tunability by adjusting multiple parameters. In the current proof-of-principle experiment, several experimental parameters (gas pressure, the blade position relative to the gas jet, and driving laser energy) are jointly adjusted to realize the tunability of LWIR wavelength. Therefore, the jumps of energy or pulse duration between different cases are not due to fluctuation or instability, but only due to the variation of multiple parameters. In order to systematically tune the LWIR wavelength and avoid jumps of energy or pulse duration, precise and reproducible control of the plasma density profile is required as mentioned in our previous theoretical paper (Ref. 10 in the manuscript).

The referee's understanding about the jump of IR pulse energy in Fig. 3 is correct. The third data point has both higher energy and longer pulse duration so that the normalized vector potentials are almost the same. The detailed XFROG data of all cases are shown in Fig. 2 and Fig. S11-S14 (Supplementary Information).

2) In order to explain the quantitative difference between experimental and simulation results, the authors notice that "the real plasma density profiles likely have longer low-density tails than the measured and simulated plasma density profiles." Does this mean that both the "online" and "offline" method are unable to measure the plasma density in such low-density tails, and why?

Limited by the signal-to-noise ratio in the plasma density measurement, the density lower than $\sim 7 \times 10^{17} \text{ cm}^{-3}$ cannot be accurately resolved, corresponding to the phase shift precision of ~ 50 mrad in our setup. Therefore, the tails of lower-density ($< 7 \times 10^{17} \text{ cm}^{-3}$) cannot be identified by our online and offline density diagnostics. Now we have clarified this in the **Discussion** section: "One possible reason is that the real plasma density profiles have long low-density tails ($< 7 \times 10^{17} \text{ cm}^{-3}$) that are below the noise level of our density diagnostics."

3) Nearly a quarter of a century after the first publication of 3D particle-in-cell simulations of laser-plasma interactions, I am still horrified by the use of only 2 particles per cell. What is the impact of such low resolution of the phase space? How can the convergence of the results be ensured? As a minor suggestion, the same temporal window should be used in Fig.4a-b and Fig.4c.f.

Indeed, resolution is an extremely important parameter for a credible simulation. The physics involved in the smallest scale has to be well resolved to give an accurate result. It is the particle number in the smallest physical volume instead of the particle number per cell that matters. For typical laser-plasma interactions, one laser/plasma wavelength

is the smallest longitudinal/transverse scale to be resolved. We followed well-accepted guideline to setup the simulation resolution. In our case, one laser wavelength ($0.8 \mu\text{m}$) is divided into 32 longitudinal grids, and one plasma wavelength ($\sim 8 \mu\text{m}$) is divided into 50 transverse grids. Even though we placed 2 particles in a cell, we actually had 160000 particles in a smallest physical volume of $8 \times 8 \times 0.8 \mu\text{m}$, which were far enough to resolve both the laser and wake evolution. **To further eliminate the referee's concern, We offer a comparative 3D PIC simulations by using 8 particles per cell and updated Fig. 4 with this simulation result.** All the laser and wake features are almost the same.

We have accepted the referee's suggestion to use the same temporal window in Fig. 4a-b and Fig. 4c-f. For comparison, we set $t = 0$ at the maximum electric field time position in both simulation and experimental results.

Reviewer #3 (Remarks to the Author):

Generating high energy mid-infrared (MIR) pulses is a topic of interest from multidisciplinary point of view. That is why most of the upcoming laser facilities are incorporating MIR lasers in their user package (See J. Phys. B: At. Mol. Opt. Phys. 50 132002 (2017) for example). In the relativistic domain accessibility to a few cycle intense pulses at several microns central wavelength would be helpful for addressing many fundamental questions. For example at present, field driven phenomena at high intensity relativistic surface physics (Phys. Rev. Lett. 110, 175001 (2013)) is fundamentally limited by the control-ability of plasma surface density gradient (L) with respect to the interaction wavelength (λ). This can be overcome when strong few cycle MIR/IR pulses would be available such that $\lambda \gg L$. In the manuscript "Photon deceleration in plasma wakes generates single-cycle, relativistic, tunable infrared pulses" the authors make a step towards generating such pulses. Thus the results are very interesting, significant and relevant. The authors use cross disciplinary techniques and simple ideas to demonstrate Strong MIR/IR pulses. The results are sound and attractive, however I have several questions that the authors need to address to improve the manuscript. Below I summarize my comments/recommendations:

1. In the abstract (last sentence) they claim demonstration of application of 'these pulses' as an in situ probe of the nonlinear wakes themselves. Elaboration on this part latter in the manuscript missed to capture my attention. They either need to remove these claim or clearly bring out where it is demonstrated in the article.

Regarding the in-situ probe of the nonlinear wakes, we have discussed in the **Discussion** section that the wake structure can be diagnosed based on the XFROG trace of the generated IR pulse. Since the IR pulse is frequency-downshifted at the beginning of a bubble and frequency-upshifted at the tail of a bubble, the Wigner spectrogram (frequency vs time) of the IR pulse can reflect the structure of the nonlinear wake. **To**

emphasize this, we have now added a sentence in the beginning of the **Discussion** section.

2. Analysis of the beam profile of the generated MIR/IR beam is missing. This kind of limits their claim of millijoules of energy. This part needs to be strengthened or clarified. Otherwise figure 3 gives the impression of extrapolation.

We did have measured the MIR/IR energy on every shot. However, as the referee said, we need to obtain the beam profile of the MIR/IR pulse to estimate the transport efficiency from generation to measurement so that the actually generated MIR/IR energy can be retrieved correctly. Due to the lack of pyroelectric cameras at hand, we were unable to measure the beam profile of the LWIR beams in our experiment. However, we have done simulations with the same parameters as the experiment. In simulations, we clearly see that all IR pulses in the range of wavelengths measured here are confined inside the wake. So the spot sizes of all IR pulses have to be less than or equal to the transverse size of the wake. This can clearly be seen in the simulation results shown in Fig. 4. Then we can estimate a lower limit of the generated IR energy by assuming the spot sizes of all IR pulses are equal to the transverse size of the wake. Using this assumption, we calculate the divergence angle and estimate the beam size at the energy meter. The energy meter intercepts only a fraction of this spot size and this is taken into account while making an estimate of the total energy conversion into the LWIR component (see Supplementary Information). To emphasize this lower limit estimation of energy, we have revised the main text in the section of **LWIR energy**: “By correcting for the transport efficiency and FWM efficiency (see Supplementary Information), the estimated mean LWIR energy (in the range 6–20 μm) generated at Gas jet 1 is **no lower than** 3.4 ± 1.1 mJ, corresponding to a conversion efficiency of **no lower than** 0.6%.”

3. In the title the authors claim and emphasize the single cycle nature of the pulse and in lines 188-191 they discuss observation of almost transform limited pulses. Few points are not brought out with clarity. Physics-wise what is the correlation between the phase properties of the driving pulse and the generated MIR/IR pulse? Why is it transform limited? What is the effect of chirp on the generated long wavelength pulses? How do the phase properties of the generated secondary long wavelength pulse depend up on the plasma stages? These are essential questions that would shed light on the control and utility of the generated pulses. I would recommend that the authors to address these points in more detail in the manuscript.

The short answer is: the generated MIR/IR pulse is carrier envelope phase (CEP) locked to the driving pulse, and the variation of the CEP will not (or weakly) affect the pulse duration of the MIR/IR pulse. This means that the MIR/IR pulse is CEP stabilized if the driving pulse is CEP stabilized. Even if the CEP is not stabilized, we can still

obtain a stabilized single-cycle MIR/IR pulse. In the experiment, the pulse duration (FWHM of the envelope intensity) of the MIR/IR pulse are measured using XFROG method. The typical pulse duration is 32.0 fs with the central wavelength of 9.4 μm , indicating a single-cycle IR pulse. This single-cycle result is not affected by the variation of CEP.

The detailed discussion regarding the referee's questions can be found in our previous theoretical paper (Ref. 10 in the manuscript).

4. In the time domain plots (figure 2 for example), it would be better to give an impression of the field and include a discussion of the CEP of the long wavelength part, and its correlation with plasma conditions and drive laser conditions. If the pulse is single cycle, such effects would be of paramount importance, hence mandatory in these types of work.

In our experiment, the XFROG method was used to measure the IR pulse information. XFROG, similar with FROG, generally measures second-order and higher-order phase of a pulse. It doesn't measure the zeroth-order phase ϕ_0 (correspond to CEP) and first-order phase ϕ_1 (correspond to absolute pulse arrival time) in the Taylor expansion of the phase. (R. Trebino, Frequency-resolved optical gating: the measurement of ultrashort laser pulses, Springer Science & Business Media, 2012). Therefore, CEP cannot be measured by XFROG. Without knowing CEP, we cannot plot the exact electric field. This is the reason why we didn't show the electric field of the MIR/IR pulse in experimental results. However, we have shown the simulated electric field in Fig. S9c in Supplementary Information.

As we stated above, if the driving laser is CEP stabilized it is possible to generate a CEP stabilized MIR/IR pulse. We agree that for future spectroscopic applications it would be desirable to have CEP locked pulses.

5. Spatiotemporal effects, which are usually part of big lasers (Appl. Phys. Lett. 104, 054103 (2014); Optics Letters 39, 4687(2014)), might play a role in the generation and observation, especially considering the propagation in an asymmetric shock density profile. A clear comment/discussion would be beneficial for the readership.

We agree that spatio-temporal effects should be carefully studied when using high peak power laser (>100 TW). However, we do not think it is a big issue in our case. The peak power of our driving laser is less than 15 TW. Also, we didn't observe significant spatio-temporal coupling effects in our 3D PIC simulations.

For scaling this scheme to higher energies using >100 TW lasers, spatiotemporal effects may affect the laser propagation and wake generation in plasmas and hence the

generation of the LWIR pulse. We have now added this sentence in the concluding paragraph of the text.

In addition, in order for the pulses to be used for subsequent experiments it would be important to know the effects of beam pointing fluctuations, and how stable the pulse energy, pulse CEP for the long wavelength pulses. I would suggest that the authors include such plots giving indication how such a source compare with existing sources.

In response to the referee's suggestion, we have added a plot of the measured IR energies for 240 different shots with fluctuation of ~31.6% (Fig. S6 in Supplementary Information). The stability of the IR pulse energy can be improved using a more stable driving laser and plasma source such as structured gas cell.

Due to the limitation of our experiment, we didn't measure the IR beam pointing fluctuation and CEP properties, which will be future work.

With these points I believe that the quality of the work would improve substantially and be suitable for publication for publication in Nature Communications.

We thank the referee for careful reading of our paper.

REVIEWERS' COMMENTS

Reviewer #2 (Remarks to the Author):

The authors' response to my concerns is almost satisfactory. The explanation that the data in Fig.3 have been obtained by tuning several parameters still leaves me some doubt on the control of the proposed scheme, but this issue may be addressed by further developments. I only recommend that the clarification given by the authors in their reply to my comments is also included in the manuscript text for the readers's sake.

I had a look also at the other referees' comments and the related authors' reply, without noticing any substantial issue. While I leave to the other referees to judge whether the reply is satisfactory, on my side I recommend publication, possibly after the above suggested final amendment.

Reviewer #3 (Remarks to the Author):

I am satisfied with the authors response and the subsequent modifications in the manuscript. In the current form the manuscript satisfies all the criteria for publication in the journal.

Response to Referees

Reviewer #2 (Remarks to the Author):

The authors' response to my concerns is almost satisfactory. The explanation that the data in Fig. 3 have been obtained by tuning several parameters still leaves me some doubt on the control of the proposed scheme, but this issue may be addressed by further developments. I only recommend that the clarification given by the authors in their reply to my comments is also included in the manuscript text for the readers' sake.

I had a look also at the other referees' comments and the related authors' reply, without noticing any substantial issue. While I leave to the other referees to judge whether the reply is satisfactory, on my side I recommend publication, possibly after the above suggested final amendment.

We have added the clarification of Fig. 3 into the section of **Wavelength tunability** as shown below, which is basically the same as our previous response to the referee.

“Wavelength tunability. The central wavelengths of the IR pulses are tuned from 3.2 to 20.0 μm by varying **several experimental parameters (gas pressure, the blade position relative to the gas jet, and driving laser energy)**. In Fig. 3, the measured IR pulse energy for different wavelength cases is plotted together with the estimated normalized vector potential a_0 of the IR pulses at the exit of the plasma structure. The detailed XFROG data for each case is shown in Supplementary Information. In all cases, the IR energy is in several millijoule level, and the a_0 is estimated to be at relativistic level ($a_0 \gtrsim 1$) at the exit of the plasma structure. **The variations in energy or pulse duration between different cases are not due to any plasma instability, but only due to the variation of multiple parameters as mentioned above. In order to systematically tune the LWIR wavelength and reduce the fluctuations in energy or pulse duration, precise and reproducible control of the plasma density profile is required¹⁰.**”

Reviewer #3 (Remarks to the Author):

I am satisfied with the authors response and the subsequent modifications in the manuscript. In the current form the manuscript satisfies all the criteria for publication in the journal.

We thank the referee for carefully reading of our paper and providing useful comments.